# Biochemical Mechanism of Fresh-Cut Lotus (*Nelumbo nucifera Gaertn*.) Root with Exogenous Melatonin Treatment by Multiomics Analysis

**DOI:** 10.3390/foods12010044

**Published:** 2022-12-22

**Authors:** Ting Min, Keyan Lu, Jinhui Chen, Lifang Niu, Qiong Lin, Yang Yi, Wenfu Hou, Youwei Ai, Hongxun Wang

**Affiliations:** 1College of Food Science & Engineering, Wuhan Polytechnic University, Wuhan 430023, China; 2Hubei Key Laboratory for Processing and Transformation of Agricultural Products, Wuhan Polytechnic University, Wuhan 430023, China; 3Key Laboratory of Agro-Products Quality and Safety Control in Storage and Transport Process, Ministry of Agriculture and Rural Affairs/Institute of Food Science and Technology, Chinese Academy of Agricultural Sciences, Beijing 100193, China; 4School Biology and Pharmaceutical Engineering, Wuhan Polytechnic University, Wuhan 430023, China

**Keywords:** fresh-cut lotus root, browning, melatonin, ROS, quality

## Abstract

Browning limits the commercial value of fresh-cut lotus root slices. Melatonin has been reported to play crucial plant roles in growth and development. However, the mechanisms in repressing the browning of fresh-cut lotuses are still unclear. In this study, fresh-cut lotus root slices were treated with melatonin, the physical signs of browning were tested, and then the selected samples (0 d, 6 d, 12 d) were used in multiomics analysis. Fresh-cut lotus root slices with a thickness of 4 mm were soaked in a 40 mmol/L melatonin solution for 10 min; then, the slices were packed in pallets and packages and stored at 10 ± 1 °C. The results show that the 40 mmol/L melatonin selected for repressing the browning of lotus roots significantly delayed the decrease in water, total soluble solid content, and Vitamin C, decreased the growth of microorganisms, enhanced total phenolic content, improved total antioxidant capacity, and decreased ·OH, H_2_O_2_, and O_2_^−^· contents. Moreover, this treatment enhanced phenylalanine ammonialyase, polyphenol oxidase, superoxide dismutase, and catalase activities and reduced peroxidase activities and soluble quinones. NnSOD (104590242), NnCAT (104609297), and some NnPOD genes showed a similar transcript accumulation pattern with enzyme activity. It can be seen from these results that exogenous melatonin accelerated an enhancement in the antioxidant system and AsA-GSH cycle system by regulating ROS-metabolism-related genes, thereby improving the capacity to withstand browning and the quality of lotus root slices. The microbiome also showed that melatonin suppressed the fertility of spoilage organisms, such as Pseudomonas, Tolumonas, Acinetobacter, Stenotrophomonas, and Proteobacteria. Metabonomics data uncovered that the metabolites of flavonoid biosynthesis, phenylpropanoid biosynthesis, tyrosine metabolism, and phenylalanine metabolism were involved in the process.

## 1. Introduction

Fresh-cut fruit and vegetables are peeled or cut into usable portions for packaging with high nutrition, the convenience of transport, and maintaining flavor [1,2,3,4]. Due to increasing consumer requirements for fresh, healthy, and safe foods, especially with the background of the global COVID-19 (Corona Virus Disease 2019) pandemic, the market for fresh-cut fruit and vegetables has grown fast [2]. Lotus root is an important edible aquatic vegetable widely distributed from Asia to all around the world. In recent years, lotus roots are increasingly sold as a fresh-cut product with crisp tissues and rich in nutrients [3,4]. During manufacturing and processing, browning, microbial growth, and nutrition loss limit the shelf life of fresh-cut lotus roots [1]. Among these factors, browning is the major restriction leading to the deterioration of the shelf-life quality of fresh-cut lotus root slices [5]. It is a challenge to extend shelf life and keep the quality of fresh-cut lotus roots during storage.

The browning of fresh-cut lotus roots is considered to be mainly caused by the enzymatic reactions of oxidoreductase. The substrates used in the enzymatic browning of fresh-cut lotus roots are phenols, among which phenylalanine ammonialyase (PAL) is the key enzyme in the synthesis of phenols, and polyphenol oxidase (PPO) and peroxidase (POD) are the main enzymes that catalyze the oxidation–reduction reaction [6]. On the other hand, several studies have indicated that the enzymatic browning of fresh-cut fruit and vegetables is closely associated with reactive oxygen species (ROS) metabolism [7]. Mechanical damage during fresh-cut processing often enhances the production of a large quantity of ROS, causing damage to cellular membranes and thus accelerating the occurrence of enzymatic reactions [8]. In fruit and vegetables, natural antioxidants (vitamins, phenolics, and carotenoids) and enzymatic antioxidant systems have an important role in ROS scavenging, including superoxide dismutase (SOD), catalase (CAT), and ascorbate peroxidase (APX) [9].

At present, many physical and chemical treatments have already shown marked potential in the preservation of fresh-cut lotus roots [10]. It was found that a 10 mmol/L oxalic acid application can effectively delay the browning of fresh-cut lotus roots. Min et al. [11] and Liu et al. [12] found that low-temperature storage, vacuum packaging, and high concentrations of carbon dioxide significantly suppress the browning and extend the shelf-life of fresh-cut lotus roots. Wang et al. [13] found that UV-C treatment could keep the quality of fresh-cut lotus roots by maintaining soluble quinone content and inactivating PPO, POD, and PAL activities. Sun et al. [5] found that exogenous H_2_S could delay fresh-cut lotus root slice browning by improving antioxidant capacities to alleviate oxidative damage. Gao et al. [4] found that 24-epibrassinolide treatment suppresses surface browning in lotus root slices by affecting phenolic accumulation (PAL and PPO) and antioxidant activity (POD, CAT, and APX). However, new methods and substances are still largely needed for the preservation of fresh food products.

Melatonin (N-acetyl-5-methoxytryptamine) is a ubiquitous hormone in nature, including both plants and animals [14]. Melatonin has been discovered in an increasing number of various fruit and vegetables, including the cherry, banana, pear, apple, and grape [15,16]. Melatonin plays a key role in regulating the circadian rhythm and photoperiod of plants [17], regulating the development of plants [18], enhancing resistance to adverse environments by reducing the content of free radicals, improving the ability of scavenging ROS, and reducing lipid peroxidation [15]. Melatonin can effectively regulate the postharvest ripening and aging processes of fruits and vegetables, delaying the decline in quality during storage [19]. Hu et al. [20] found that melatonin could delay the softening and deterioration of banana quality. Zheng et al. [21] found that the treatment of 100 µmol/L melatonin could improve the antioxidant capacity and ROS scavenging capacity of pears. Aghdam and Fard [22] and Liu et al. [16] found that melatonin treatment reduced H_2_O_2_ accumulation and increased total phenol and antioxidant activities in strawberries. Exogenous melatonin treatment can inhibit the fresh-cut fruit and vegetable browning of pears [21] and broccoli [23], but the effects of exogenous melatonin application on the browning and quality of fresh-cut lotus roots have not been reported.

Melatonin is famous for meliorating sleep problems and whitening the skin in humans and promoting seed germination, delaying leaf senescence, and increasing oxidation resistance in plants [18], while its effect on the quality of fresh-cut fruit and vegetables is still unclear. In this study, the effect of melatonin on the quality of fresh-cut lotus roots was examined by multiple omics, and changes were found in microbial growth, physiological change, oxidation pathways, and phenylpropanoid metabolism during lotus root storage treated with melatonin. Our results provide a reference for the application of exogenous melatonin for the storage and preservation of fresh-cut lotus roots.

## 2. Materials and Methods

### 2.1. Material and Treatments

The pretreatment method was performed according to the methods of Zhou et al. [8] Lotus roots (“Elian 5”) were harvested from a commercial wetland (Hankou North, Wuhan) in July and then transported quickly to the laboratory and precooled at 4 °C for 24 h. Seriously damaged lotus roots were removed during cleaning and peeling. Lotus roots with uniform size and color were selected as the experimental material and cut into 4 mm thick slices with a slicing machine. The cut lotus root samples were randomly divided into two groups, and the following treatments were applied.

In the melatonin treatment group (Melatonin), the prepared slices were immediately immersed in melatonin solution (40 mmol/L, through gradient filter, data not shown) for 10 min. After soaking, excess water was drained off, and the slices were packed in pallets and packages. The slices were stored at 10 ± 1 °C. For the control group (control and mock), the prepared slices were immediately immersed in ethanol solution (control) and tap water (mock) for 10 min. and then drained off and packed in pallets and packages.

The packages were 200 × 280 mm and made of polyethylene. The pallets were 180 × 120 × 25 mm and made of polypropylene. All experiments were repeated three separate times. The frozen samples were stored at −80 °C for further research.

### 2.2. Appearance Quality

The appearance quality was recorded with a camera (Canon, EOS550D, Tokyo, Japanese).

The determination of color was performed according to the method of Liu [2]. The values of L*, a*, and b* of the full surface of the fresh-cut lotus root were measured with a colorimeter (JZ-300, Shenzhen Jinzhun Instrument Equipment Co., Ltd., Shenzhen, China).

### 2.3. Weight Loss, Total Soluble Solid Content, and Vitamin C (V_C_) Content Evaluated

The weight loss rate was evaluated by the weighing method [24].

Total soluble solid content was analyzed according to the method of Liu et al. [16]. A total of 10 g of lotus root tissue was ground in a mortar and filtered. A sample (2–3 mL) was filtered and measured with a portable refractometer (LQ80T, Speedway Electronic Technology Co., Ltd., Guangzhou, China).

The V_C_ content was measured according to the method of Ali et al. [25]. A total of 5 g of lotus root with 50 mL of water was ground in a mortar and incubated in an ice bath for 10 min. Then, the solution was centrifuged at 10,000× *g* for 10 min at 4 °C, and the supernatant was collected as a crude VC extract. A total of 10 mL of the supernatant was titrated with the calibrated 2,6-dichlorophenol-indophenol solution until it turned reddish and did not fade for 15 s. V_C_ content was expressed as mg per 100 g of fresh weight (mg/100 g).

### 2.4. Total Bacterial Count, Total Phenol Content, and the Activity of PAL, SOD, and CAT Measurement

Total Bacterial Count was determined according to the method of Wang [13]. The lotus root tissue was treated with 0.85% stroke-physiological saline solution (225 mL). After preparing the serial dilutions, the cells were isolated onto NA medium at 37 °C for 48 h, and the total bacterial count was determined. The bacterial count was expressed as log10 CFU/g.

The extraction and measurement of phenols were carried out according to our previous report [6]. Total phenol content was assayed by Folin–Ciocalteu, and the results were expressed as gallic acid equivalent per 100 g of fresh weight (mg/100 g).

PAL was extracted and its activity was analyzed as described in a previous study [6]. The PAL enzyme activity unit (U) was defined spectrophotometrically as a change in absorbance of 0.1 at 290 nm per minute per gram of fresh weight.

The extraction and analysis of SOD activity were performed as described using a Superoxide Dismutase Kit. Five grams of lotus root tissue was homogenized in 20 mL of phosphate buffer (0.05 M, pH = 7.0) on ice, and the solution was then centrifuged at 10,000× *g* at 4 °C for 10 min. The supernatant was collected as a crude SOD extraction. The reaction mixture consisted of reagent 1 (1 mL), 0.05 mL of supernatant (0.05 mL of distilled water for the control), reagent 2 (0.1 mL), reagent 3 (0.1 mL), and reagent 4 (0.1 mL). The mixture was incubated at 37 °C for 40 min, and chromogenic agent (2 mL) was added to the above mixture and incubated for 10 min at 37 °C. The absorbance value of the sample at 550 nm was measured. SOD activity was defined as follows: a unit of SOD activity (U) corresponded to the content of SOD when the inhibition rate of SOD reduced to 50%.

The extraction and analysis of CAT activity was analyzed according to the method of Jannatizadeh [26]. Lotus tissue (5 g) was mixed in 20 mL of phosphoric buffer solution (0.05 M, pH = 7.0) on ice, and the solution was then centrifuged at 10,000× *g* at 4 °C for 10 min. The supernatant was collected as a crude CAT extract. Then, 100 μL of crude CAT extract was added to 2.9 mL of H_2_O_2_ solution (20 mmol/L), and the changes in the absorbance of the mixture at 405 nm were measured. An enzyme activity unit (U) was defined spectrophotometrically as a change in absorbance of 0.01 at 405 nm per minute.

### 2.5. Gas Headspace Analysis, Soluble Quinone Content, MDA Content and ROS Production

The headspace gas concentrations in the packages were measured according to Wang et al. [13]. Headspace gas concentrations were determined with an O_2_/CO_2_ analyzer (Checkpoint3, Mocon Co., Ltd., Shanghai, China.

Total soluble quinone content was analyzed based on Wang’s study [13]. Lotus root tissue (5 g) was homogenized in anhydrous alcohol (20 mL) and centrifuged at 10,000× *g* for 10 min at 4 °C. The supernatant was collected and measured at 437 mm by a spectrophotometer. The content of soluble quinone was expressed as OD_437nm_/g.

The MDA content was determined as previously described by Liu et al. [2] with slight modification. 3 g of Lotus root tissue was homogenizing in 10% trichloroacetic acid (15 mL), followed by centrifugation at 4 °C (10,000 × *g*, 20 min). After that, a similar volume of the supernatant and thiobarbituric acid (*w*/*v*, 0.67%) was added and boiled for 20 min. After cooling, the above centrifugation step was repeated. The MDA content was expressed as mmol/g.

The O_2_^−^· production rate was analyzed according to the method of Gao [24]. A total of 5 g of root tissue was homogenized in 5 mL of extraction buffer solution (1 mmol/L EDTA, 0.3% Triton X-100, and 2% PVP) in an ice mortar, and the solution was centrifuged at 12,000× *g* at 4 °C for 20 min. The supernatant samples were collected as a crude O_2_^−^· extract. Next, the reaction mixture was mixed with 1 mL of crude O_2_^−^· extract, 1 mL of phosphate buffer (50 mmol/L, pH = 7.8), and 1.0 mL of hydroxylamine hydrochloride (1 mmol/L). The sample was static-settled at 25 °C for 1 h, and 1 mL of sulfanilic acid (1.7 mmol/L) and 7 mmol/L α-naphthylamine were added to the above mixture. Then, the samples were centrifuged at 12,000× *g* for 5 min at 4 °C, and the absorbance of the supernatant was measured at 530 nm. The O_2_^−^· production rate was expressed as nmol per minute per g of fresh weight (nmol/(min·g)).

The ·OH content was tested as described with a Hydroxyl Radical Kit (Nan Jing Jiancheng Bioengineering Institute, Nanjing, China). Lotus root tissue (5 g) was mixed in 20 mL of absolute ethyl alcohol in an ice mortar, and the solution was centrifuged at 10,000× *g* at 4 °C for 10 min. The supernatant was collected and diluted 50 times as a crude ·OH extract. The samples were mixed with reaction mixture and treated following the instruction book; then, the mixture was incubated for 1 min at 37 °C, and 2 mL of chromogenic agent was added to the above mixture. The absorbance of the sample at 550 nm was measured. The ·OH production rate was expressed as nmol per minute per g of fresh weight (nmol/(min·g)).

The H_2_O_2_ content was analyzed following the explanatory memorandum of Hydrogen Peroxide Kit (Jiancheng Bioengineering Institute, Nanjing, China). A total of 2 g of lotus root tissue was washed in 18 mL of NaCl solution (0.85%) in an ice mortar, and the mixture was centrifuged at 10,000× *g* at 4 °C for 10 min. The supernatant was collected as a crude H_2_O_2_ extract. The reaction mixture consisted of 1 mL of reagent 1, 0.1 mL of crude H_2_O_2_ extract (0.1 mL of distilled water was in the blank tube, and 0.1 mL of 163 mmol/L standard solution was in the standard tube), and 1 mL of reagent 2. The absorbance of the sample at 405 nm was measured. H_2_O_2_ content was expressed as mmol per g of fresh weight (mmol/g).

### 2.6. Multiomics Analyses of Exogenous Melatonin Treatment

RNA sequencing of fresh-cut lotus root slices in control and Melatonin groups (stored for 0 d, 6 d, and 12 d) was performed on the DNBSEQ platform by staff at Beijing Genome Institute (BGI) (Shenzhen, China). Transcript abundances were expressed as FPKM (expected number of Fragments Per Kilobase of transcript sequence per Millions base pairs sequenced). Three biological replicates for each sample time and treatment were conducted.

Microbiota colony total DNA was extracted by using E.Z.N.A.^®^ soil DNA kit (Omega Bio-tek, Norcross, GA, USA). The NEXTFLEX Rapid DNA-SEQ Kit was used for library construction, and then Illumina sequencing was performed on the company’s Miseq PE300 platform.

Metabonomics were performed using advanced mass spectrometer Xevo G2-XS QTOF (Waters, Manchester, UK), Progenesis QI (version 2.2) (Waters, Manchester, UK), and metaX, wherein identification was based on KEGG database. The differentially expressed metabolites were identified by determining the following parameters: variable importance in projection (VIP) values in multivariate PLS-DA model, fold change (FC), and q-value of univariate analysis. The filtering rules were: (1) VIP values were equal or greater than 1; (2) fold changes were equal to or greater than 1.2 or less than or equal to 0.8; and (3) q-values were less than 0.05. All three had to be met for an ion to be considered a differential ion. Metabolic pathway analysis was based on the KEGG database.

### 2.7. Quantitative Real-Time PCR Validation of RNA-Seq Data

A total of 10 DEGs involved in ROS metabolism were further identified using qRT-PCR. GenBank number and ID of the DEGs are listed as follows: 104611642 (XM_010278785.1); 104603071 (XM_010267042.2); 104599952 (XM_010262707.2); 104594740 (XM_010255193.2); 104608049 (XM_010273911.2); 104590242 (XM_010248818.2); 104600124 (XM_010262965.2); 104592473 (XM_010251858.2); 104591893 (XM_010251015.1); 104602250 (XM_010265863.2). The reference gene selected in this study was β-Actin (XM_010243420.2), and analysis method was 2^−ΔΔCT^. Primers are listed in Appendix A. The data are shown as mean ± standard deviation.

### 2.8. Statistical Analysis

All samples were analyzed with three replications, and the results are expressed as the mean ± standard deviation. Analysis of variance (ANOVA) was carried out with IBM SPSS Statistics 25 (SPSS Inc., Chicago, IL, USA). Differences between means were assessed by Dunnett’s test, with differences considered significant at *p* < 0.05.

## 3. Results and Discussion

### 3.1. Exogenous Melatonin Treatment on Fresh-Cut Lotus Root Slice Retard Browning

Browning is one of the major factors limiting fresh-cut lotus root slice quality and shelf life. We sprayed lotus root slices with exogenous melatonin and its solvent, and as shown in Figure 1A, the initial surface color of the lotus root slices was bright white, but the color of control slices was pale brown on 6 d and changed to a dark color on 12 d of storage, which indicated that melatonin and the solvent group delayed the change in appearance, with melatonin showing better effects than the solvent group (ethanol solution). The weight loss rate in fruit and vegetables directly reflect the quality and senescence of fruit and vegetables during storage [27]. The weight loss rate of melatonin-treated lotus root slices increased more slowly than the ddH_2_O-treated group (Figure 1B). Lightness (L*) decreased with storage time, while the L* value of the solvent-treated group (Control) was slightly lower than the melatonin-treated group, indicating that melatonin and ethanol treatment could prevent the decrease in lightness (Figure 1C). a* and b* increased during storage, and compared with those of the control, the a* value and b* of fresh-cut lotus roots in the melatonin-treated group were significantly decreased (Figure 1D,E). Therefore, melatonin reduced the decrease in L* and inhibited the increase in a* and b* on the surface of fresh-cut lotus roots.

### 3.2. Exogenous Melatonin Treatment Inhibited the Oxidation of Phenolic Compounds

Soluble solids are another parameter that directly reflects the quality and senescence of fruit and vegetables during storage [27]. The total soluble solids of the melatonin-treated group were significantly higher than that of the mock group during storage while it was the same as that of the control group (Figure 2A). These results suggest that melatonin and ethanol solution can effectively inhibit water loss and total soluble solids reduction during storage. MDA content is an important index to evaluate the tolerance of plants to stress. MDA content was obviously increased after the alcohol treatment while melatonin dramatically suppressed this process (Figure 2B).

Phenolic substances are not only important substrates for enzymatic browning but also important antioxidants for fruit and vegetables [2]. Throughout storage, the total phenolic content of the melatonin group was significantly higher than that of the control and mock groups (Figure 2C), indicating that the remaining phenols were higher because of the protection of melatonin. Similar findings have been reported for pomegranates [28], mangoes [29], and fresh-cut pears [21]. Soluble quinone is the main product of the browning reaction [13]. Soluble quinone content significantly increased throughout storage compared with the initial concentration on 0 d in the control group (Figure 2D). Nevertheless, melatonin application markedly inhibited the increase in soluble quinones compared to the mock and control conditions, and the ethanol solution also depressed soluble quinones production. After 12 days of treatment, soluble quinones were substantively (0.42-fold) lower in melatonin-treated slices than water-treated slices. This result is consistent with the appearance quality, browning degree, and color difference, suggesting that melatonin can inhibit the synthesis of soluble quinone and delay the browning of fresh-cut lotus roots [4,10].

PAL is an important enzyme in the synthesis of phenolic substances. As shown in Figure 2E, throughout storage, the PAL activity of the melatonin group was significantly higher than that of the control and mock groups, which was consistent with the trend in total phenol content. In addition, this result suggests that the high PAL activity of lotus root slices treated with melatonin resulted in high phenol content, high DPPH scavenging capacity, and high V_C_ (also called ascorbic acid) content, all of which are valuable for the protection of membrane integrity, which is beneficial for delaying the browning and aging of fresh-cut lotus roots. Similar findings have been reported for the pomegranate fruit [26,28] and broccoli [23]. V*_C_* is not only an important indicator for evaluating the nutritional value of fruit and vegetables but also an important biologically active substance [30]. During the whole storage process, the V*_C_* content decreased. The V*_C_* bleed rate was significantly slower after the melatonin treatment. The V*_C_* content of the mock group decreased from 41.24 mg/100 g FW to 21.14 mg/100 g FW, a decrease of 48.74%, similar to the control group, while the V*_C_* content of the melatonin group decreased from 41.24 mg/100 g FW to 28.46 mg/100 g FW, a decrease of only 30.99% (Figure 2F). Additionally, the V*_C_* content of the melatonin group was significantly lower than that of the control group throughout storage, indicating that exogenous melatonin could delay the decrease in V*_C_* content in fresh-cut lotus roots. Zhu [23] found similar results after treating fresh-cut broccoli with melatonin, which may be related to the high antioxidant activity of melatonin.

### 3.3. Melatonin Application Increased the Antioxidant Capacity of Fresh-Cut Lotus Root Slices

In the present research, a decreased concentration of ROS coincided with reduced browning in the melatonin compared with the control condition, which indicated that melatonin application possibly resulted in significantly reduced browning in the treated lotus slices by acting as an antioxidant to reduce the overproduction of ROS. Similar findings have been reported in pears [21]. Our present study indicated that fresh-cut lotus root slices have relatively strong respiration during early storage that gradually decreases in the later period and that melatonin can effectively delay the respiration of fresh-cut lotus roots during storage. In this study, our results show that the concentrations of CO_2_ and O_2_ sharply increased and decreased, respectively, during the initial 6 d of storage and then did not change significantly in the later stages of storage. The CO_2_ concentration of the melatonin group was lower than that of the control group during the storage period, but the concentration of O_2_ of the melatonin group was higher than that of the control group (Figure 3A,B). The respiration of fresh-cut lotus root slices changed dramatically after melatonin application.

The mechanical damage induced by cutting generally leads to the overproduction of ROS, and this overproduction is considered harmful to the storage of lotus root slices [4,10]. Reactive oxygen species (ROS) have been implicated as the cause of oxidative damage to membrane lipids, one of the most convincing mechanisms of apoptosis and natural senescence in plants, such as superoxide anion (O_2_^−^·), hydrogen peroxide (H_2_O_2_), and hydroxyl radical (·OH) [29]. The production rate of ·OH was significantly increased during the initial 2 days of storage in nontreated slices and decreased in the later stages of storage. The production rate of ·OH in the melatonin group was significantly lower than that in the mock group, except on the 10th day (Figure 3C). The O_2_^−^· content was significantly and linearly increased during the early stage of storage but decreased from day 8 in the control slices (Figure 3A). The O_2_^−^· production rate of the melatonin group was significantly lower than that of the mock group (*p* < 0.05). The O_2_^−^· production rate was lower in melatonin-treated slices than in mock and control slices (Figure 3D).

The content of H_2_O_2_, another important ROS substance, was significantly increased during the initial 4 days of storage in nontreated slices and decreased in the later stages of storage. However, melatonin application markedly inhibited H_2_O_2_ production compared to the control from the fourth day (Figure 3E). On the 12th day, H_2_O_2_ content was 0.12-fold lower in melatonin-treated slices than in control slices. SOD converts O_2_^−^· into H_2_O_2_ and O_2_ through a disproportionation reaction, eliminating the damage of superoxide anions to cells [24]. CAT mainly acts on high concentrations of H_2_O_2_ and transforms H2O2 into H_2_O and O_2_. The CAT activity of lotus root slices gradually increased in early storage but decreased on the 12th day regardless of treatment (Figure 3F). Nevertheless, CAT activity was substantively higher in melatonin-treated slices than in control and mock slices. SOD activity increased during storage in the control group after 6 days, while SOD activity was stimulated in melatonin-treated slices compared with control slices (Figure 3G) (*p* < 0.05).

Exogenous melatonin application improved the storage quality of fresh-cut lotus root slices by inhibiting respiration. H_2_O_2_ was significantly lower in melatonin-treated slices than in control slices, probably due to higher CAT activity, which plays a key role in H_2_O_2_ decomposition. In contrast, O_2_^−^· was significantly lower in the melatonin group than in the control, possibly due to higher SOD activity. Melatonin eliminated excess ROS by increasing SOD and CAT enzyme activity and then delayed the occurrence of browning.

### 3.4. Melatonin Treatment Reduced the Microorganism Proliferation on the Fresh-Cut Lotus Roots

The analysis of the changes in appearance and weight loss clearly shows that melatonin inhibited the surface browning of fresh-cut lotus root slices during storage. The inhibition of browning by melatonin was also found in fresh-cut pears and litchi fruit [21,31]. We performed a Wien analysis of the number of common and unique species and found that the melatonin-treated group had a lower number of common and unique species. The rapid growth of microorganisms often leads to the prompt deterioration of fresh-cut products during storage; therefore, total bacterial count (TBC) is an imperative indicator that can be used to determine the shelf-life of fresh-cut products. Fresh-cut produce with high microorganism levels is generally considered unsafe for consumers [31]. Our present study showed that melatonin had the ability to reduce the microbe-mediated spoilage of fresh-cut lotus root slices. In this study, the TBC was significantly increased at the early stage of the storage period and slightly decreased at the later stage compared with the beginning of the study in melatonin-treated and nontreated control slices; however, the increase in the TBC of melatonin-treated slices was significantly lower compared with that of control slices (Figure 4A,B).

The species level was also clustered, and norank-c-Cyanobacteria (cyanobacteria), Pantoea (pantobacter), Pseudomonas (Pseudomonas), norank-f-Mitochondria, Tolumonas, Acinetobacter (Acinetobacter), and Stenotrophomonas (oligoaeromonas) were enriched in the mock group. In the melatonin-treated group, there were five species of dominant bacteria in different storage periods, including norank-c-Cyanobacteria, Pantoea, Pseudomonas, norank-f-Mitochondria, and Leuconostoc. During the storage process of the fresh-cut lotus roots, the microorganisms showed different growth rules. The richness of norank-c-Cyanobacteria decreased during the storage of the fresh-cut lotus roots, and the decreasing rate of the mock group was higher than that of the melatonin group. The contents of Pseudomonas, Tolumonas, and Acinetobacter were very low (less than 1%) on the 0th day of fresh-cut lotus root storage, and the richness of the mock group increased gradually during the later storage. Meanwhile, the Pseudomonas richness of the melatonin-treated group was only more than 1% on the 12th day, while the richness of Tolumonas and Acinetobacter was less than 1% during the whole storage period. The richness of Pantoea increased during storage. On the 6th day, the richness of Pantoea in the mock group was higher than that in the melatonin-treated group, and that in the mock group was lower than that in the melatonin-treated group on the 12th day; the richness of norank-f-Mitochondria decreased during storage; the content of Stenotrophomonas was more than 1% only in the mock group after 12 days of storage; and the content of Leuconostoc was more than 1% only in the melatonin-treated group after 12 days of storage (Figure 4C). In summary, Pseudomonas, Tolumonas, Acinetobacter, Pantoea, and Stenotrophomonas may lead to the metamorphic lotus root increasing during storage, and melatonin treatment may delay the quality deterioration of fresh-cut lotus roots by inhibiting the growth and reproduction of Pseudomonas, Tolumonas, Acinetobacter, and Stenotrophomonas.

### 3.5. Transcriptomic Profiling of the Effect of Exogenous Melatonin Treatment on Fresh-Cut Lotus Roots

In order to reveal the molecular difference between lotus roots treated with or without exogenous melatonin during storage, fresh-cut lotus roots with three storage periods (0 days, 6 days, and 12 days) were selected in this experiment. A total of 27 samples were tested using RNA-seq technology. In total, the nine libraries generated between 25.84 and 21.62 million raw reads, and the total number of clean reads per library ranged from 21.19 to 22.77 million. We used HISAT to compare clean reads to the reference genome: ‘China Antique’ reference genome 2 (http://www.ncbi.nlm.nih.gov/genome/genomes/14095 (accessed on 10 January 2020)) using Bowite2. The comparison results are counted as shown in Appendix A. Between 81.09 and 96.62% of the short clean reads were aligned against the ‘China Antique’ reference genome 2. Between 85.57 and 98.21% of the clean reads were uniquely aligned against the reference genome.

A WGCNA related to genes expressed in the successive developmental stages was performed as shown in Appendix A. We found that 937 genes were correlated with melatonin treatment, and 238 were correlated with the untreated sample. Genes were clustered in accordance with the storage time of the melatonin treatment as shown in Appendix A. The principal component analysis (PCA) results of this project are shown in Figure 5A and Appendix A. In the analysis of this transcriptome, it is convenient to find that the mock and control groups were discrete while the melatonin-treated group showed relative centralization, especially the samples collected at 12 d. We further compared these samples, and as shown in Figure 5B, we found 331 specific genes expressed in the mock group and 648 specific genes expressed after the melatonin treatment. As shown in Figure 5C,D, genes expressed in the untreated group were enriched in the external encapsulating structure organization, cell wall organization, extracellular region heme binding, terapyrrole binding, and drug catabolic process, and a large number of genes were enriched in oxidoreductase activity, while the specific genes expressed after the melatonin treatment were enriched in pentose and glucuronate interconversions, phenylpropanoid biosynthesis, plant–pathogen interaction, and plant hormone signal transduction.

Taller plants have evolved extremely advanced systems to protect cells from oxidative damage, by being equipped with essential non-enzymatic antioxidants and ROS-scavenging enzymes (i.e., SOD, CAT) for scavenging superoxide radicals and H_2_O_2_ [32]. To further confirm the function of exogenous melatonin in the storage quality of fresh-cut lotus roots, candidate differentially expressed genes related to antioxidants and ROS scavenging were isolated based on the transcriptome data. We further built a heat map of ROS-related genes in Figure 5E. Under melatonin treatment, the transcript levels of the assayed *NnPOD* genes exhibited two different kinds of changing trends over time. Among them, the transcripts of 12 *NnPOD* genes increased in control and decreased when treated with exogenous melatonin throughout the experimental period. For example, the transcript levels of 104605488 and 104611642 were elevated to 1425.9 (2-fold higher than the initial point) and 160.93 (162-fold higher than the initial point), respectively, in the control group and were significantly suppressed in the melatonin-treated group at day 12. The expression profiles of another 3 *NnPOD* genes were down-regulated to various degrees in all slices, which were consistently lower in the melatonin-treated group than in control. In comparison to the initial point, the transcript level of 104608049 decreased, respectively, to 1/3 and 1/24 in the control and melatonin-treated groups after 12 days of storage (Figure 5E and Appendix A). These findings indicate that melatonin suppressed the expression of *NnPOD* genes, and POD activity might be regulated by multiple gene family members. In our study, it was found that the SOD and CAT activity of fresh-cut lotus roots was substantively higher in the melatonin-treated group compared with the control group (Figure 3F,G). *NnSOD* (104590242) showed a similar transcript accumulation pattern with observed changes in the corresponding enzyme activity, which was significantly higher in the melatonin-treated slices than control slices (Figure 3F,G). The expression of *NnCAT* (104609297) distinctly decreased with the storage duration in all slices (Figure 5E and Appendix A). However, melatonin treatment up-regulated *NnCAT* expression and slowed the rate of decline. The transcriptional level changed in the RNA-seq was further confirmed by qRT-qPCR, subsequently. Ten DEGs related to ROS metabolism were chosen to design a primer (Appendix A). The expression levels of the selected genes were tested, and the results are shown in Appendix A. According to qRT-PCR POD (104590242), CAT (104600124) and DHAR (104592473) were up-regulated during the melatonin treatment. In contrast, POD (104611642, 104599952, and 104594740), CHS (104591893), and ESPS (104602250) were down-regulated by the melatonin treatment, and these findings are consistent with RNA-seq results.

### 3.6. Metabolomics Changes of the Fresh-Cut Lotus Roots Treated with Exogenous Melatonin

Metabolomics analysis helps us screen out specific metabolites in certain biological processes and clarify the metabolic processes and mechanisms of changes in organisms. We performed the principal component analysis (PCA) and found the same tendency with the transcriptome (Appendix A). A total of 3580 differential secondary metabolites were detected. We enriched these compounds in the KEEG pathway with time series, as shown in Appendix A. After 6 days, the up-regulated metabolites of the mock group were enriched in processes including sesquiterpenoid and triterpenoid biosynthesis, tyrosine metabolism, phenylpropanoid biosynthesis, phenylalanine metabolism, limonene and pinene degradation, and isoquinoline alkaloid biosynthesis, while after 12 days, the metabolites of the mock group were enriched in almost the same processes. After melatonin treatment, the up-regulated metabolites were enriched in processes including phenylalanine metabolism, tyrosine metabolism, tryptophan metabolism, and sesquiterpenoid and triterpenoid biosynthesis, while the amounts of secondary metabolites reduced remarkably. The down-regulated metabolites of the mock group were enriched in processes including tyrosine metabolism, phenylpropanoid biosynthesis, phenylalanine metabolism, pentose and glucuronate interconversions, isoquinoline alkaloid biosynthesis, and flavonoid biosynthesis. After melatonin treatment, the down-regulated metabolites were enriched in processes including valine, leucine, and isoleucine biosynthesis, pentose and glucuronate interconversions, C5-branched dibasic acid metabolism, ascorbate and aldarate metabolism, purine metabolism, phenylpripanoid biosynthesis, biosynthesis of amino acids and ascorbate and aldarate metabolism. The amounts of secondary metabolites are also remarkably reduced.

We further compared the metabolomics changes of the fresh-cut lotus roots treated with exogenous melatonin after 12 days with the mock group. As shown in Figure 6A, we found flavonoid biosynthesis, phenylpropanoid biosynthesis, tyrosine metabolism and phenylalanine metabolism. The content of compounds is shown in Figure 6B, and the detailed information can be found in Appendix A. 2,4-Dihydroxyhept-2-enedioic acid, cinnamaldehyde, phenylacetylglycine, (R)-2-Methylimino-1-phenylpropan-1-ol, 4-Hydroxyphenylacetylglycine, 2,4-Dihydroxyhept-2-enedioic acid, and Phenylic acid were induced after melatonin treatment, while p-Coumaraldehyde, Homoeriodictyol chalcone, 5-O-Caffeoylshikimic acid, Phenylacetaldehyde, Phenylacetic acid, L-Normetanephrine, 2-(3,4-Dihydroxyphenyl)ethylamine, and 5-Oxopent-3-ene-1,2,5-tricarboxylate were only induced in the mock group.

## 4. Conclusions

In conclusion, with the 40 mmol/L melatonin treatment, the quality of fresh-cut lotus root browning was effectively controlled and maintained. After melatonin treatment, the O^2−^, H_2_O_2_, and ·OH in fresh-cut lotus root slices were significantly scavenged, and the growth of microorganisms was obviously inhibited. The total phenolic contents, TTS, V_C_, and oxidation resistance ability were enhanced by melatonin. In addition, our research also showed that melatonin treatment increased PAL, PPO, SOD, and CAT activities and that exogenous-melatonin-treated slices showed lower POD activities and lower soluble quinone contents than control slices. *NnSOD* (104590242), *NnCAT* (104609297), and some *NnPOD* genes showed a similar transcript accumulation pattern with enzyme activity changes. It can be seen from these results that exogenous melatonin accelerated an enhancement in the antioxidant system through regulating ROS-metabolism-related genes, thereby improving the capacity to withstand browning and the quality of fresh-cut lotus root slices. An integrated multiomics framework of the fresh-cut lotus root response to melatonin treatment reduced the microbial reproduction and depressed the physiological activity of fresh-cut lotus roots from the transcriptional level to the metabolic level to keep the slices fresh. We suggest that exogenous melatonin treatment is a promising approach to control the cut surface browning and quality of lotus root slices, which is still a problem for fresh-cut lotus roots.

## Figures and Tables

**Figure 1 foods-12-00044-f001:**
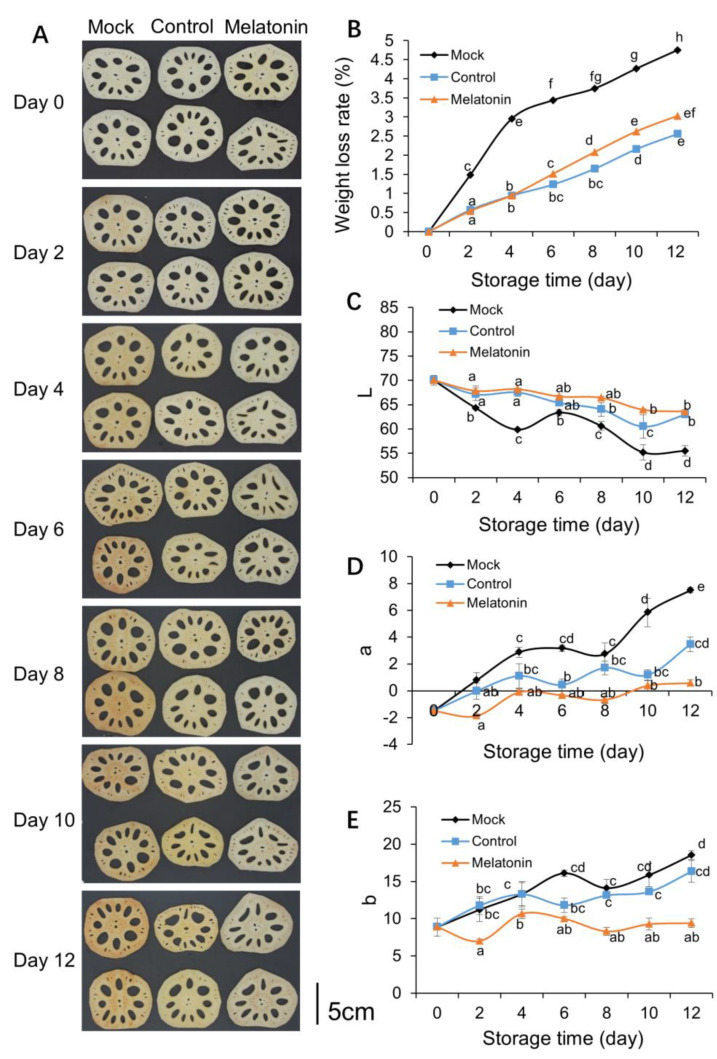
Effect of melatonin on the appearance, water loss, and color (**A**–**E**) of fresh-cut lotus roots (‘E Lian 5’). Fresh-cut lotus roots were treated with tap water (control, left triangle) or 40 mmol/L melatonin solution (melatonin, circles). Error bars represent SEs from three biological replicates. Different letters represent significant differences at *p* < 0.05 level.

**Figure 2 foods-12-00044-f002:**
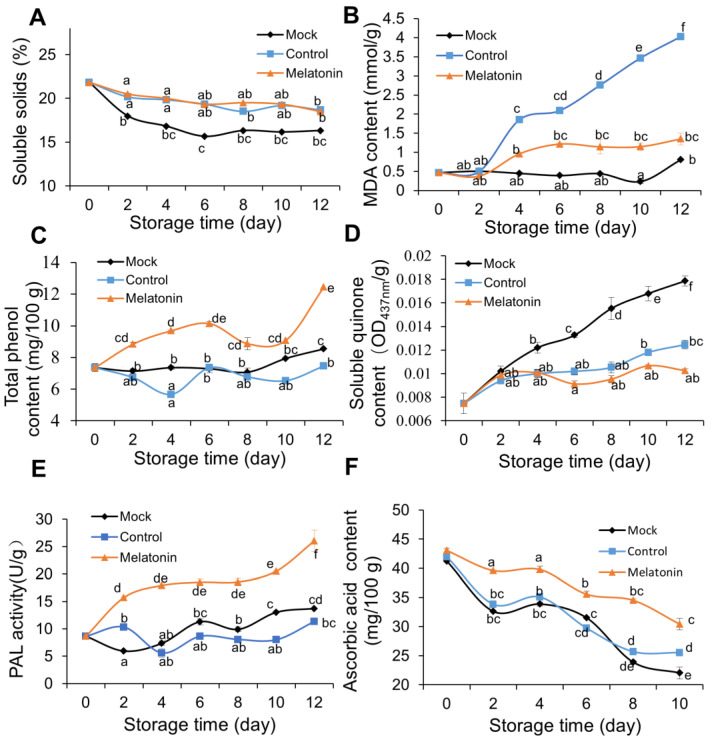
Effect of melatonin on the soluble solids (**A**), MDA content (**B**), total phenol content (**C**), soluble quinone gas concentration (**D**), PAL activity (**E**), and vitamin C content (**F**) of fresh-cut lotus roots (‘E Lian 5’). Fresh-cut lotus roots were treated with tap water (control, left triangle) or 40 mmol/L melatonin solution (MELATONIN, circles). Error bars represent SEs from three biological replicates. Different letters represent significant differences at *p* < 0.05 level.

**Figure 3 foods-12-00044-f003:**
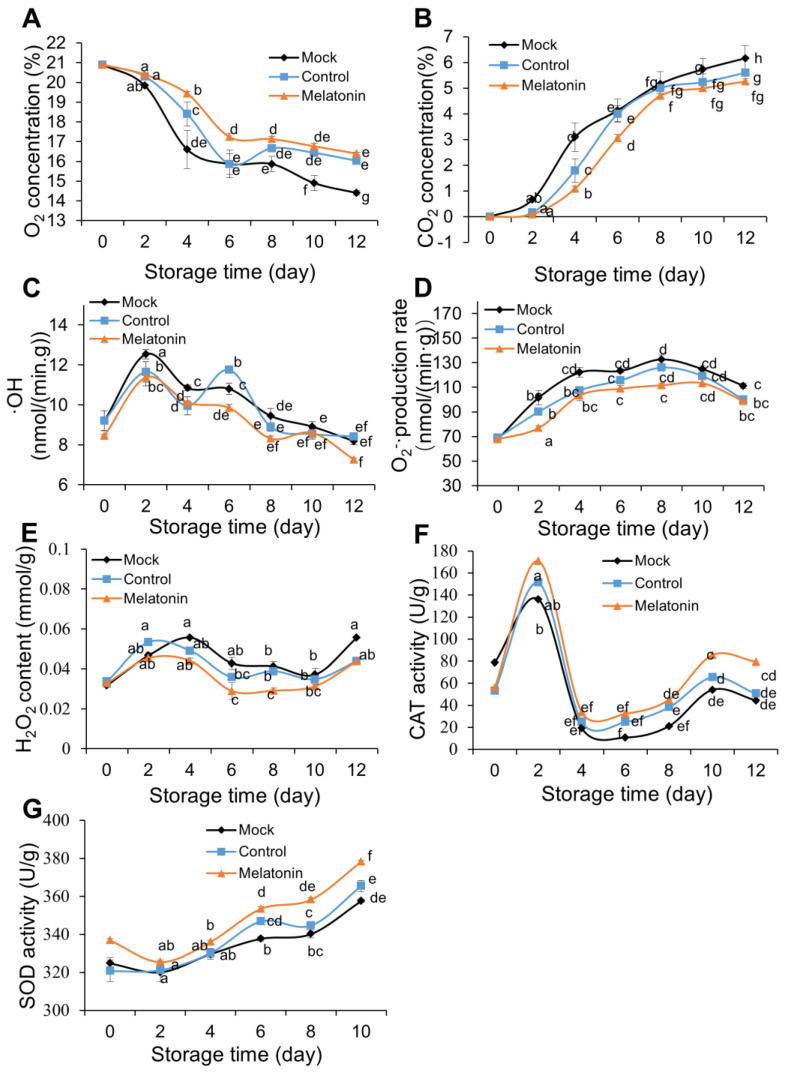
Effect of melatonin on gas headspace change (**A**,**B**), ROS (**C**–**E**) and SOD activity (**F**), and CAT activity (**G**) of fresh-cut lotus roots (‘E Lian 5’). Fresh-cut lotus root was treated with tap water (Mock), ethanol (Control), or 40 mmol/L melatonin solution (melatonin). Error bars represent SEs from three biological replicates. Different letters represent significant differences at *p* < 0.05 level.

**Figure 4 foods-12-00044-f004:**
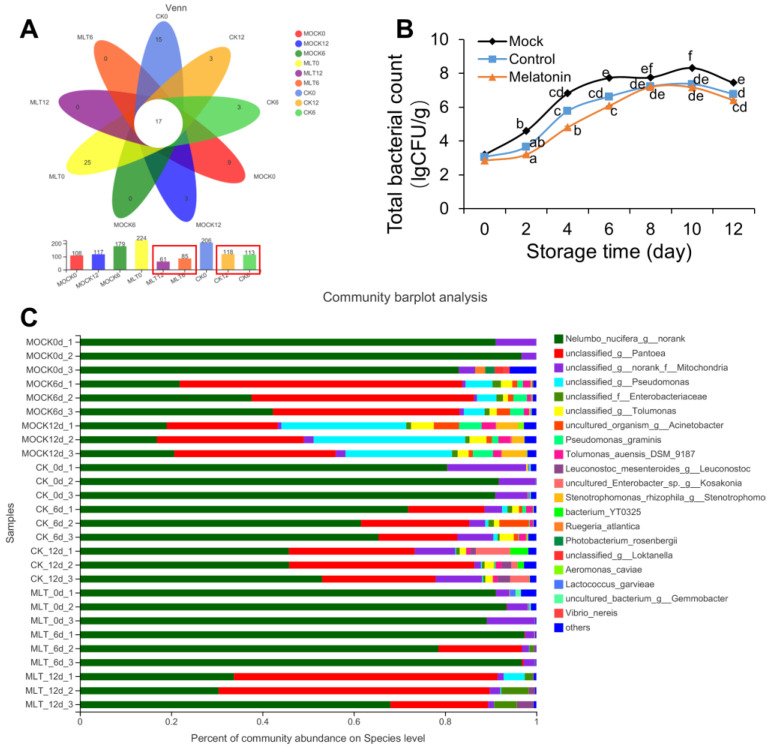
Effect of melatonin on common and unique species. (**A**) Venn analysis of the number of common and unique species in different samples. (**B**) Total bacterial count of samples after treatment. (**C**) Microbial community composition of fresh-cut lotus roots. Error bars represent SEs from three biological replicates. Different letters represent significant differences at *p* < 0.05 level.

**Figure 5 foods-12-00044-f005:**
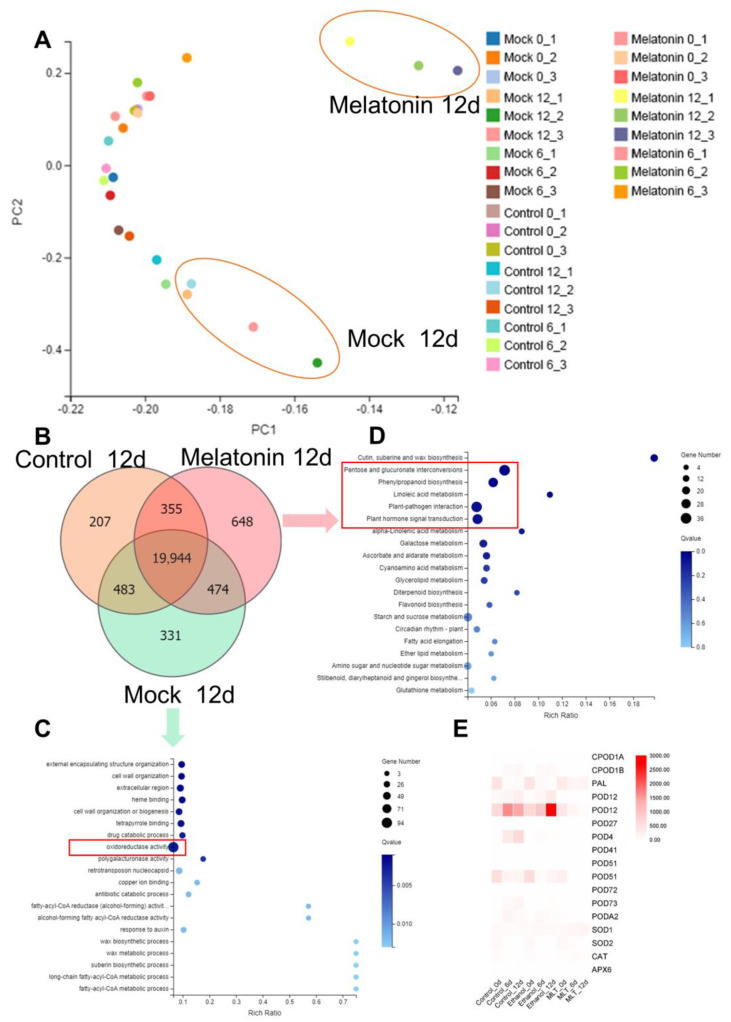
The expression profiles of fresh-cut lotus roots treated with melatonin solution. (**A**) PCA, principal component analysis. (**B**) Venn diagrams of groups in fresh-cut lotus root (Mock), fresh-cut lotus root treated with ethanol solution (CK), and fresh-cut lotus root treated with melatonin dissolved in ethanol solution (Melatonin); samples were all collected after 12 days. (**C**,**D**) KEEG pathway enrichment of specific expression genes in different groups. (**E**) Heat map of the expression profiles of ROS-related genes in the transcriptome.

**Figure 6 foods-12-00044-f006:**
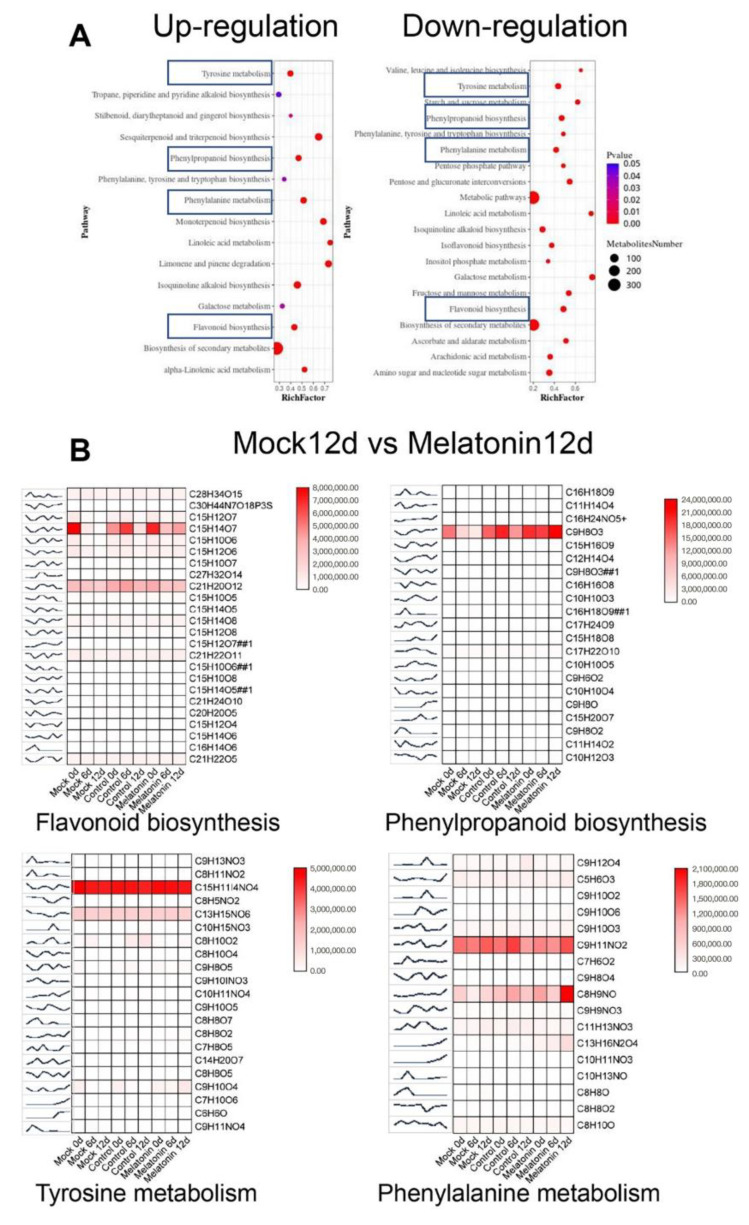
Metabolomics changes of the fresh-cut lotus roots treated with exogenous melatonin. KEEG pathway enrichment of compounds in different treatments after treatment for 12 days (**A**). Differential ion cluster analysis of compounds enriched in selected pathway (**B**).

## Data Availability

Data are contained within the article.

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
