# Peer review of "Biochemical Mechanism of Fresh-Cut Lotus (Nelumbo nucifera Gaertn.) Root with Exogenous Melatonin Treatment by Multiomics Analysis"

_foods, 2022, doi:10.3390/foods12010044_

Round 1

Reviewer 1 Report

Manuscript foods-2048758

The manuscript describes the use of melatonin to reduce the browning process of fresh-cut lotus. The manuscript includes a huge experimental work and combines several disciplines. The description of results and metabolomics are well-connected y allow an advanced interpretation of the biochemical mechanisms involved in the brownish mitigation. Unfortunately, several figures include typo sizes that are too small and hardly readable. Despite understanding that the problems are mainly derived from the programmes used, authors should do their best to improve the presentations; otherwise, readers could give up the reading before ending it. It would be a pity. In addition, a revision for small typo or syntaxis would be convenient.  

Title: Currently, it does not properly catch the idea behind the research objectives. A suggestion could be:

“Browning mitigation of fresh-cut lotus (Nelumbo nucifera Gaertn.) root by exogenous melatonin treatment. Study of the involved biochemical mechanisms by multi-omics analysis”.  

However, authors should feel free to choose any other option they consider more appropriate. Also, consider using italics for the scientific name of lotus.  

Abstract. It should include at least a sentence (or modify some already in text) to present the experimental design. It could help for a more straightforward interpretation of the following description of results.  

L39. The citation in the text could not be adequate.

L226 Please correct the syntax error.

L246 Please complete the version and provider.

L283. Please melatonin did not increase the total phenol. The remaining phenols were higher because of the protection of melatonin.

L303-313. The use of VC and Vc creates confusion.

Figure 4A and Figure 4C should be improved. Particularly, legends are hardly read.

L431. Please revise

L432-433 Please revise

Figure 5 C, D, and E are hardly read. Please improve them.

Figure 6. idem previous comment. In general, several of the figures (or sections) should be improved. The actual typo sizes are too small.

L536. Please, revise.

Reviewer 2 Report

The authors must homogeneously handle the units in the document.

The description of the methods must be in units and not in words.

Document with minimal observations is attached

Pag 3. Line 107. What are the characteristics of the container? It only mentions that they are made of polyethylene: high or low density? what is the thickness? What is the permeability of the container?
